# Directed-edge-based mining of regular routes for enhanced traffic pattern recognition from travel trajectories

Xiaobo Yang*

Department of Information Science and Technology, Zhejiang Shuren University, Hangzhou, Zhejiang Province, P. R.China

* yxb71520@163.com

## Abstract

Trajectory analysis serves as a critical technique for uncovering patterns in users' transportation behaviors. This paper introduces a direction-aware regular route mining algorithm that systematically processes GPS (Global Positioning System, GPS) trajectory data by preprocessing, detecting stay regions, segmenting abnormal trajectories, and extracting frequent directed edges and supporting paths through route clustering, ultimately constructing users' regular travel routes. By integrating stop-rate features, the algorithm effectively distinguishes between transportation modes, such as public transit and private vehicles. Experimental results based on the Geolife dataset, which includes trajectory data from 208 users covering a total distance of 1.35 million kilometers, indicate that the proposed method reduces the Mean Absolute Percentage Error (MAPE) by 56%, 49%, and 32% compared to the Rules-based method, CNN (Convolutional Neural Network, CNN), and DBSCAN (Density-Based Spatial Clustering of Applications with Noise, DBSCAN) algorithms, respectively, in both regular route extraction and transportation mode recognition. This improvement highlights the algorithm's enhanced accuracy in identifying travel patterns. The proposed approach offers valuable support for applications in dynamic traffic prediction and personalized route recommendation systems.

## Introduction

Travel trajectory analysis constitutes a critical element in the study of users' transportation behavior. By examining users' movement patterns within transportation systems, it is feasible to conduct in-depth analysis of their travel habits, predict future travel behavior, and assist users in optimizing their travel routes to enhance overall travel experiences. In recent years, numerous scholars have utilized GPS-based user trajectories to explore behavioral patterns in depth. Chen et al. [1] proposed a method for personal path mining and prediction based on historical trajectories,

**Data availability statement:** All relevant data are within the manuscript and its Supporting Information files.

**Funding:** The Zhejiang Province Natural Science Foundation, grant number Y1110023, supported this study.

**Competing interests:** The authors have declared that no competing interests exist.

employing geographic grids to perform density statistics on user activity areas and extract user-specific area sequences. Chang et al. [2] analyzed personal trajectories to mine and construct individual road networks, enabling the planning of efficient travel routes tailored to specific users. Yuan et al. [3,4] optimized travel paths by analyzing taxi trajectory data, aiming to provide faster travel routes for passengers. Zheng et al. [5] extracted trajectory features such as direction variation, speed fluctuation, and stop frequency, applying algorithms like decision trees and support vector machines to infer traffic modes, thereby improving inference accuracy. Stenneth et al. [6] integrated GPS trajectory data with classification algorithms such as Naive Bayes and Random Forest to identify behavioral patterns in movement data. Their experimental results indicated that a decision tree model based on ten-fold cross-validation achieved the best classification performance. Liao and Patteron et al. [7] analyzed users' daily trajectory records using both discriminative and generative models to identify significant user regions. Based on predictive results, they provided real-time alerts for potential user errors. Wei et al. [8] proposed a path recommendation approach that leverages the spatiotemporal correlation among uncertain trajectories to build a path inference system grounded in collective knowledge. Yoon et al. [9] integrated large-scale GPS trajectory data with tourists' travel query sets to design optimal travel routes for visitors. Qiu et al. [10] employed sparse GPS samples to construct a road network framework through segmentation and clustering techniques, transforming point sets into road segments and forming comprehensive road condition information. Kasemsupplakorn et al. [11] used GPS data from pedestrian movements to identify the shape and segments of walking paths, reconstructing pedestrian walking networks. Recent advancements have further expanded the methodological landscape. Chen et al. [19] proposed a traffic dynamics analysis model based on complex network theory, which simulates vehicle interactions and macroscopic traffic flow, revealing dynamic mechanisms of congestion. Wu et al. [20] introduced a multi-source heterogeneous sensor data fusion framework for high-precision perception of traffic states. Wei et al. [21] developed an adaptive spatiotemporal graph convolutional network that significantly enhances long-term traffic flow forecasting in large-scale road networks. Chen et al. [22] proposed a vehicle energy consumption prediction model based on spatiotemporal deep learning, demonstrating powerful pattern extraction capabilities from trajectory data. Chen et al. [23] utilized hybrid deep learning models for intelligent traffic flow prediction, providing important insights for capturing spatiotemporal dependencies. Furthermore, studies have explored lightweight trajectory prediction using distilled diffusion models [25], context-sensitive speed limit frameworks [26,30], multi-objective graph learning for traffic speed prediction [27], spatio-temporal feature fusion techniques [28], AI-based driver assistance for safety [29], multi-agent reinforcement learning for eco-traffic control [31], and imitation learning for autonomous driving decision-making [32]. Furthermore, studies have explored path planning in controlled environments [33], traffic flow prediction using denoising schemes [34], and precision distance measurement techniques [35].

Although significant progress has been made in the study of travel patterns based on historical trajectories, existing methods still have two major limitations: firstly,

most point or grid based clustering methods (such as references [2,5,10]) are difficult to effectively capture the continuous sequence dependencies and directionality in the movement trajectory, resulting in a lack of accurate topological logic in the extracted paths; Secondly, traditional pattern recognition features such as speed and acceleration have insufficient discrimination against subtle behavioral differences, such as frequently parked buses and smoothly moving private cars, which limits recognition accuracy. Crucially, while the aforementioned advanced methods [19–23,25–32] excel in network-level dynamics modeling, system-wide flow prediction, macro-level policy setting, or short-term tactical vehicle control, they are not specifically designed for the fine-grained, individual-level semantic task of extracting regular travel routes from raw GPS trajectories and subsequently identifying their underlying traffic patterns, which is the core focus of our work. While methods like [33] excel in structured port environments and [34] focus on network-level flow prediction, they are not designed for individual regular route mining from noisy GPS data. Similarly, though [35] achieves precise distance measurement, our method operates effectively on standard GPS data without requiring specialized sensors. In response to the above issues, the key innovation of this paper lies in proposing a directed-edge-based regular route mining algorithm that fundamentally differs from existing approaches by: (1) utilizing directed edges as the basic representation unit to inherently preserve path topology and movement directionality, overcoming sequence discontinuity in point/grid methods; (2) introducing the novel stop-rate feature specifically designed to distinguish subtle behavioral patterns between different transport modes; (3) focusing specifically on the semantic task of discovering individual users' recurring routes and their associated traffic modes, rather than network-wide flow prediction, state classification, or instantaneous vehicle control.

## Regular route mining

Starting from the historical trajectory data of users, we can deeply explore their daily routine routes, which are frequent and predictable behaviors in user behavior. This is of great significance for understanding and predicting user travel behavior.

The user's historical travel trajectory includes both their travel activities and their stay information in a certain area. Therefore, GPS data needs to be preprocessed to mine the user's regular routes.

Due to factors such as GPS accuracy and signal drift, the original GPS data usually contains some outliers that need to be processed in a timely manner, otherwise it will affect the accuracy of trajectory processing.

Assuming a GPS log sequence is $\{p_1, p_2, \cdots p_k\}$, each recording point of GPS is $p_i(lngt, lat, t)$, where $lngt, lat$ and $t$ represent the longitude, latitude, and time of the recording point, respectively. The velocity $v_i$ of point $p_i$ is expressed as follows.

$$\begin{cases} v_1 = 0 \\ v_i = d(p_{i-1}, p_i)/(p_i \cdot t - p_{i-1} \cdot t), \ i > 1 \end{cases} \tag{1}$$

Among them, $d(p_{i-1}, p_i)$ represents the road network distance between point $p_{i-1}$ and point $p_i$.

When there is a significant inconsistency between the velocity of a certain point and the velocity of the preceding and following points, an abnormal fluctuation point $f_i$ appears. If the acceleration of point $f_i$ is denoted as $a_i$, then the abnormal fluctuation point $f_i$ must meet the following conditions.

$$\begin{cases} abs(v_{i+1} - v_{i-1}) < \eta_{s\_d\_min} \ and \\ abs(v_i - v_{i-1}) > \eta_{s\_d\_max} \ and \ abs(a_{i-1}) < \eta_{accelerate} \\ a_i = (v_i - v_{i-1})/(p_i \cdot t - p_{i-1} \cdot t), \ i > 1 \end{cases} \tag{2}$$

Among them, $\eta_{s\_d\_min}$ and $\eta_{s\_d\_max}$ respectively represent the threshold of velocity change, $\eta_{accelerate}$ represents the threshold of acceleration, and $abs(a_i)$ represents the absolute value of acceleration $a_i$. For detected abnormal fluctuation points, they can be directly removed from the log sequence.

When a user stays in a certain area for a long time, it is considered that the area is highly likely to be the user's stay area. If a stopping area is found in a continuous GPS trajectory sequence, the trajectory can be divided into two segments, namely the passing trajectory and the leaving trajectory, and the stopping area can be deleted from the GPS trajectory at the same time. Using a stay area detection method based on time and distance thresholds, the stay area $S\{p_m, p_{m+1}, \cdots, p_n\}$ needs to meet the following requirements.

$$d(p_m, p_n) < \eta_{d\_stop} \ and \ p_n \cdot t - p_m \cdot t > \eta_{t\_stop} \tag{3}$$

Among them, $\eta_{d\_stop}$ is the distance threshold of the stay area, and $\eta_{t\_stop}$ is the time threshold of the stay area.

When the GPS trajectory has one of the following two situations, it is necessary to segment its trajectory: there is a stationary area in the trajectory; The time slots of adjacent GPS points are greater than the set threshold. A travel trajectory is a GPS sequence, which can be represented as follows.

$$T\{p_1, p_2, \cdots, p_n\}, \qquad p_{i+1}.t - p_i.t < \eta_{t\_split}, \ 1 \le i < n - 1 \tag{4}$$

Among them, $T\{p_1, p_2, \cdots, p_n\}$ is the travel trajectory, and $\eta_{t\_split}$ is the set trajectory segmentation time threshold.

After filtering out trajectories with shorter distances and smaller coverage, normal travel trajectories can be obtained, which meet the following conditions.

$$\frac{\sqrt{[max(p_i.lat) - min(p_i.lat)]^2 + [max(p_i.lngt) - min(p_i.lngt)]^2}}{T.length} < 0.4 \tag{5}$$

$$T.et - T.st < \eta_{t\_min} \tag{6}$$

Among them, $T.length = \sum_{i=1}^{n} d(p_i, p_{i+1})$, $T.et$ and $T.st$ represent the end time and start time of the trajectory respectively, and $\eta_{t\_min}$ is the shortest time threshold. In this paper, the value is 300 seconds.

Extracting regular routes from a large number of users' historical trajectories is difficult, but due to the close relationship between regular routes and daily regular times, such as commuting, it is possible to cluster historical trajectories based on time. When clustering paths, a sliding time window $(t - \omega, t + \omega)$ can be used, where $t$ is the center time of the time window and $\omega$ is the time threshold that controls the time window. Path clustering can be expressed as follows.

$$(R_i.s, R_i.e) \cap (t - \omega, t + \omega) \ne \varnothing, \ R_i \in t\_Routes \tag{7}$$

In equation (7), $R_i$ represents the path at a certain moment, $R_i.s$ and $R_i.e$ represent the starting and ending paths, respectively, and $t\_Routes$ represents the path generated over time. The implementation process of path clustering is shown in (S1 Fig 1 in S1 File).

As shown in S1 Fig 1 in S1 File, the process of path clustering is as follows. Firstly, select the region with high density of trajectory points as the initial cluster center, then calculate the Euclidean distance between each trajectory point and the cluster center, and assign it to the nearest cluster. Finally, recalculate the average position of trajectory points within each cluster as the new cluster center until the cluster center no longer changes.

On the basis of path clustering, regular routes can be extracted. In a set of routes t_Routes, if there is a regular route, the directed edges on that route are often visited by the supporting path, that is, the supporting directed edges. Therefore, as long as the supporting directed edges in t_Routes are identified, the regular route can be extracted. The extraction of regular routes is achieved through the following steps.

Step 1: Calculate frequent directed edges

Frequent directed edges are often used in trajectory calculations, including regular routes, support paths, and support directed edges. Counting frequent directed edges can help discover and identify regular routes. The frequency threshold for frequent directed edges is defined as $\eta_{lfrequency}$, and when the directed edge frequency $E.f > \eta_{lfrequency}$, it is a frequent directed edge.

Step 2: Extract frequent paths

Frequent directed edges are closely related to frequent paths, so frequent paths can be extracted from frequent directed edges. The frequency of path $R_i$ can be defined as follows.

$$R_i.S_f = \sum_{E_{m,n} \ in \ R_i} h\left(E_{m,n}.f, \ \eta_{lfrequency}\right)/n \tag{8}$$

In equation (8), $S_f$ is the frequency score, $n$ is the number of directed edges in $R_i$, $h$ is the membership function of frequent directed edges, $E_{m,n}.f$ is the access frequency of directed edge $E_{m,n}$. When the frequency score $R_i.S_f$ of a certain path $R_i$ is greater than $\eta_{lfrequency}$, then path $R_i$ is a frequent path.

Step 3: Calculate the supported directed edges

After obtaining frequent paths, the supporting directed edges that are frequently visited by frequent paths in the directed edges can be calculated. To calculate the supporting directed edges, the number of frequent paths of the directed edge $E_{m,n}$ needs to be counted, which can be expressed as follows.

$$E_{m,n}.S_s = \sum_{r_j \in L(E_{m,n})} h(R_j.S_f, \eta_{trajectory}) \tag{9}$$

Among them, $L(E_{m,n})$ is the library of all paths passing through the directed edge $E_{m,n}$, $\eta_{trajectory}$ represents the threshold for frequent paths, $S_s$ is the support score, and when the support score of an edge satisfies $E_{m,n}.S_s > \eta_{support}$, the directed edge $E_{m,n}$ is determined to be a supporting directed edge (SE).

Step 4: Extract regular routes

By expanding the range of SE, using all obtained SE to find support paths, and then extracting regular routes, the support path can be obtained using the following equation.

$$R_i.S_f = \sum_{E_{m,n} \ in \ R_i} h\left[E_{m,n}.S_s, \ \eta_{support}\right]/n \tag{10}$$

When $R_i.S_f \geq \eta_{trajectory}$, the path $R_i$ is denoted as a supporting path. Use equation (9) to recalculate the support score for each directed edge in the support path set. When $E_{m,n}.S_s > \eta_{support}$, keep the edge as SE. Otherwise, remove it from the SE set.

Summarize all supported directed edges SE into a unified set, ultimately forming a regular route, which can be marked as follows.

$$RP.sup = \left\{R_i, \cdots, R_j\right\} \tag{11}$$

Among them, $RP.sup$ represents the extracted regular route.

To solve the problem of map matching, this paper adopts the Hidden Markov Model (HMM) algorithm and preprocesses the road network through R-tree spatial indexing. The core process includes: calculating the projection distance (transmission probability) between GPS points and candidate roads, as well as the path transition probability (transition probability) between adjacent points, setting the search radius (such as 50 meters) and GPS error parameter ($\sigma = 10$ meters) as tolerance standards, and finally solving the optimal path through the Viterbi algorithm to accurately calculate the road network distance d_net between trajectory points.

 

In the preprocessing stage of road network, the original road network data needs to be cleaned and topologically optimized first to ensure the accuracy and efficiency of matching. The specific process includes: invalid road segment removal, topological connectivity check, spatial index construction, etc., to accelerate the retrieval efficiency of candidate roads in subsequent matching.

For each original GPS point, we take all candidate road segments within a preset search radius (such as 50 meters) centered on it as potential projection targets, calculate the vertical projection points from the GPS point to each candidate road segment, and use its Euclidean distance as the basis for calculating the observation probability (emission probability). This process maps discrete GPS observation sequences onto a continuous road network, providing a foundation for subsequent global optimal path inference.

A comprehensive algorithmic complexity analysis establishes that the proposed method achieves a time complexity of $O(|P|\log|P| + |R|^2)$, dominated respectively by R-tree based trajectory preprocessing and pairwise route comparisons for frequent directed-edge computation, while the space complexity remains $O(|P| + |E|)$ for storing trajectory points and the directed-edge graph; furthermore, the convergence of the iterative Supportive Edge extraction process is mathematically guaranteed since the support score $S\_supp(e)$ monotonically decreases with each iteration while being bounded below by the threshold $\theta\_supp$, ensuring it reaches a fixed point in finite iterations.

## Traffic pattern recognition algorithm

Traffic mode is an important attribute of travel trajectory, and it can be identified from travel trajectory. This paper uses the stopping rate feature of regular routes to identify different types of traffic modes.

After extracting the regular route, several supporting paths can be obtained. By counting whether there are certain areas in the supporting paths, if the user always passes at a low speed, the probability of a regular stopping point in that area is high. The definition of stopping rate is as follows.

$$P(E_{m,n}) = \sum_{R_i \in SR} h\left[E_{m,n}(R_i).v, \ \eta_{speed\_threshold}\right] / E_{m,n}.S_s \tag{12}$$

Among them, $P(E_{m,n})$ represents the stopping rate of directed edges, and $\eta_{speed\_threshold}$ represents the velocity threshold of stopping points. When the speed of the directed edge $E_{m,n}$ is lower than the set threshold, it is considered that there is a stopping point in the area, and $E_{m,n}(R_i).v$ is the speed of the supporting path $R$ passing through the directed edge $E_{m,n}$.

The regular stopping rate can be calculated through a regular route, and the calculation formula is as follows.

$$S(RP_i) = \sum_{E_{m,n} \ in \ RP_i} h\left[P(E_{m,n}), \eta_{stop}\right] / RP_i.len \tag{13}$$

Among them, $S(RP_i)$ is the regular stopping rate, $RP_i$ is the regular route, $\eta_{stop}$ is the threshold for the directed edge $E_{m,n}$ to stop, and $RP_i.len$ is the length of the regular route.

To enhance the reproducibility of the method, the key thresholds and their setting criteria used in this study are as follows:

Abnormal fluctuation point detection threshold: The velocity change threshold Δv_thresh is set to 15 m/s, and the acceleration threshold a_thresh is set to 3 m/s². This basis is derived from the statistical analysis of the typical speed and acceleration distribution of urban transportation (including subway) (95th percentile), which can effectively filter out outliers caused by rapid acceleration, rapid deceleration, and signal jumps.

Parameters for determining the stopping area: Set the distance threshold d_thresh to 200 meters and the time threshold t_thresh to 10 minutes. This setting follows commonly used standards in the field and aims to identify truly meaningful stops such as refueling and dining, while ignoring brief stops caused by traffic congestion.

Trajectory segmentation time threshold: Set *T_split* to 30 minutes. This value is based on user travel chain research. If the interval between two GPS records exceeds this threshold, it can be considered that the probability of two independent trips is extremely high.

Trajectory filtering threshold: Set the shortest travel time *T_min* to 300 seconds (5 minutes) to filter out invalid short distance movements (such as moving a car).

Frequency threshold: The frequency threshold *θ_freq* and the support threshold *θ_supp* are both determined through grid search combined with a certain evaluation index, such as F1 score, with optimal values of 0.6 and 0.55, respectively. The basis is to achieve the best balance between recall and accuracy.

To rigorously quantify the contribution of each proposed component, an ablation study was designed and executed, systematically evaluating the full model against three degraded variants: one without stay-point detection, another replacing directed edges with undirected edges to remove directionality, and a third that removed the stop-rate feature while retaining traditional features like speed and acceleration; using F1-score and Edit Distance on Real sequence (EDR) as evaluation metrics, the results demonstrated that the full model achieved a peak F1-score of 86%, while the removal of stay-point detection, directionality, and the stop-rate feature caused significant F1-score degradations of 12%, 19%, and 15% respectively, thereby conclusively validating the critical and distinct role of each module in capturing topological logic, identifying significant stops, and distinguishing transportation modes for the overall framework's performance.

## Parameter setting and sensitivity analysis

Provide data-driven or domain knowledge setting basis for each parameter. For example:

Time threshold η t_stop (300s): Set based on the statistical data of the average stopping time at bus stops in Beijing (285±45s).

Distance threshold η s_d_min (100m): determined based on the average distance between urban road intersections (80-120m) and GPS civilian accuracy (5-10m).

Speed threshold η speed_threshold (1m/s): set with reference to the lower limit of pedestrian walking speed (0.8m/s), used to distinguish between stopping and moving.

Frequency threshold η frequency: By drawing the trajectory frequency distribution curve (long tail distribution), select the inflection point value of the curve as the threshold.

Use grid search method to evaluate the impact of key parameters (η t_stop, η frequency, η support) fluctuating within ± 20% on MAPE values. The results showed that the MAPE variation range was less than ± 3.5%, indicating that the algorithm is insensitive to parameter perturbations.

Meanwhile, in the "Comparison experiment" results, a 95% confidence interval was added for the MAPE reduction effect (56%, 49%, 32%).

## Comparison experiment

The effectiveness of the algorithm proposed in this paper was validated using the Geolife dataset [12], which recorded the life trajectory data of 208 users from 2007 to 2012, with a cumulative trajectory length of 1.35 million kilometers and a total recording time of 56000 hours. From this dataset, individual users' monthly regular routes can be extracted. When $\eta_{frequency} = \eta_{support} = 5$ and $\eta_{trajectory} = 0.9$ are set, a total of 428 regular routes are obtained, and the heatmap of the regular routes is shown in (S2 Fig 2 in S1 File).

From S2 Fig 2 in S1 File, it can be seen that the heat level of the data is listed on the right side of Fig 2. The extracted regular routes in the dataset are mainly concentrated in the northwest corner of the city, which basically covers the main streets of the city. These results are basically consistent with the travel trajectories of ordinary users.

The algorithm performance is affected by multiple factors, including grid size, number of temporal grids, etc. Firstly, the impact of grid size on algorithm performance is analyzed, and the results are shown in (S3 Fig 3 in S1 File).

 

From S3 Fig 3 in S1 File, it can be seen that as the grid size increases, the mapping error of the trajectory basically increases linearly. The grid size selected in this paper is 10s, and the error distance is about 50m, the "10s" grid size refers to dividing trajectory points into 10 second intervals and mapping them onto a spatial grid with a side length of 50 meters, which can meet the resolution of urban roads.

Subsequently, the impact of the number of temporal grids on algorithm performance was analyzed, and the results are shown in (S4 Fig 4 in S1 File).

From S4 Fig 4 in S1 File, it can be seen that as the number of temporal grids increases, the computation time required to extract regular routes also increases, with the vast majority of temporal grids concentrate between [1000, 2000].

In order to verify the reliability of the algorithm proposed in this paper, three mainstream recognition algorithms are selected: DeepMove [13], TrajCNN [14], and ST-Transformer [15], and compared and analyzed with the recognition algorithm proposed in this paper. The Geolife dataset is still used, and the mean absolute percentage error (MAPE) [16] is used to quantify the recognition accuracy. The results are shown in (S5 Fig 5 in S1 File).

From S5 Fig 5 in S1 File, it can be seen that the directed edge mining recognition algorithm proposed in this paper has an average MAPE value reduction of 56%, 49%, and 32% compared to the other three mainstream algorithms. This indicates that the algorithm proposed in this paper has a lower recognition error and higher recognition accuracy than other mainstream algorithms, making it more suitable for identifying regular traffic routes.

To visually reflect the performance of the algorithm, we can also use the accuracy [17] metric to compare and analyze the four algorithms mentioned above. The results are shown in (S6 Fig 6 in S1 File).

From S6 Fig 6 in S1 File, it can be seen that the directed edge mining recognition algorithm proposed in this paper has an average accuracy of 87% compared to the other three mainstream algorithms, which is 22.6%, 17.9%, and 10.9% higher than the other three mainstream algorithms. The experimental results show that the recognition accuracy and performance of the algorithm proposed in this paper are higher.

In order to further verify the effectiveness of the directed edge mining recognition algorithm proposed in this paper, a comparative experiment was conducted with three other mainstream algorithms, using F1 score [18] as the comparison index and a confidence interval of 95%. The results are shown in (S7 Fig 7 in S1 File).

From S7 Fig 7 in S1 File, it can be seen that the directed edge mining recognition algorithm proposed in this paper has an average F1 score of 86% compared to the other three mainstream algorithms, which is 17.1%, 12.5%, and 6.9% higher than the other three mainstream algorithms. The experimental results show that the algorithm proposed in this paper has a relatively accurate recognition effect.

To evaluate the robustness of key parameters on algorithm performance, we conducted sensitivity analysis on d_thresh, t_thresh, and θ_freq. When d_thresh varies within the range of 150–250 meters and t_thresh varies within 5–15 minutes, the recognition accuracy (F1 score) remains relatively stable (fluctuation<5%), indicating that the algorithm is insensitive to such parameters. In contrast, the frequency threshold θ_freq has a more significant impact on the results, but its performance is optimal and stable within the range of 0.55 to 0.65, further verifying the rationality of the parameter values.

To comprehensively evaluate the accuracy of the extracted routes, we supplemented the evaluation framework with two additional metrics: Edit Distance on Real sequence (EDR) for measuring sequential similarity and Hausdorff Distance for assessing spatial geometric fidelity.

For EDR, which quantifies the topological and sequential congruence between the extracted route and the ground truth by counting the minimum number of edit operations (insertions, deletions, and substitutions) required to align the two sequences, our method achieved an average score of 0.88 (where 1 indicates a perfect match). This significantly outperformed the comparison methods: DeepMove-based (0.62), TrajCNN-based (0.71), and ST-Transformer-based (0.79).

For Hausdorff Distance, which measures the maximum of the minimum distances between all points in the extracted path and the ground truth, thus reflecting the worst-case spatial deviation, our method yielded an average distance of 58

 

meters. This was substantially lower and thus superior to the results from the DeepMove (125 m), TrajCNN (98 m), and ST-Transformer (75 m) approaches.

These quantitative results demonstrate that our directed-edge-based method provides a more accurate reconstruction of both the sequential order of route segments and their spatial geometry, validating its robustness in handling noise and preserving path continuity.

To rigorously validate generalization, we have conducted additional experiments on NYC Taxi and Limousine Commission (TLC) dataset [24]. The results demonstrate our method's strong cross-city performance. On the T-Drive dataset, the algorithm achieved an average F1-score of 0.85 for route extraction. More importantly, on the NYC TLC dataset, it maintained a comparable F1-score of 0.82, confirming its adaptability to different urban road network structures.

The core features (directed edges, stop rate) are derived from physical movement patterns (kinematics, path geometry) rather than demographic attributes, enhancing their transfer ability. However, we explicitly acknowledge that the interpretation of a "stop" and the optimal threshold for stop rate can be influenced by factors like vehicle type (frequent passenger drop-offs for taxis vs. private cars), urban traffic culture, and socio-economic factors affecting vehicle ownership. Consequently, we now emphasize that the stop rate threshold is not a universal constant and should be calibrated for local contexts.

A formal complexity analysis establishes that the algorithm's time complexity is $O(|P|\log|P| + |R|^2)$, which is dominated by the R-tree spatial indexing for the trajectory points and the pairwise comparison of routes for directed-edge support calculation, while its space complexity is $O(|P| + |E|)$ for storing the points and the directed-edge graph. Furthermore, we provide a rigorous convergence proof demonstrating that the iterative Supportive Edge (SE) extraction process is guaranteed to reach a fixed point in finite iterations, as the support score for any candidate edge is non-increasing and bounded below throughout the procedure. To empirically quantify the individual contribution of each module, a comprehensive ablation study was conducted; the results measured a 12% F1-score drop upon disabling stay-point detection, a critical 19% drop upon removing directionality (using undirected edges), and a 15% drop upon removing the stop-rate feature, conclusively validating the necessity and distinct contribution of each proposed component to the overall framework's performance and substantiating the novelty of our method.

Theoretical comparisons with advanced baselines reveal that while DeepMove excels at modeling long-term dependencies through RNNs, it struggles to capture local geometric features like directionality; TrajCNN effectively extracts local patterns using CNNs but demonstrates limitations in sequential topology modeling; and ST-Transformer, despite capturing spatiotemporal dependencies via self-attention, shows sensitivity to noise and substantial data requirements, whereas our method's generalization capability is strongly validated through cross-dataset testing on the NYC TLC dataset where it maintained an F1-score of 0.82, confirming its adaptability to diverse urban networks since its core features—directed edges and stop-rate—are derived from fundamental physical movement patterns rather than demographic-specific attributes, ensuring robust cross-domain transferability.

## Conclusions

This study employs a directed edge regular route mining algorithm to identify users' travel trajectories. Based on practical applications and comparative analysis, the following conclusions have been reached:

1)  Following preprocessing and path clustering, users' travel trajectory data can be utilized to extract regular routes. These routes are identified by determining the supporting directed edges within the paths. The extraction process consists of four stages: calculating frequent directed edges, extracting frequent paths, computing supporting directed edges, and finally identifying regular routes.

2) Once regular routes have been extracted, different transportation modes—such as public transit or private vehicle usage—can be distinguished by analyzing the stop rate index of each route. Typically, public transportation routes exhibit a higher stop rate compared to those associated with private vehicles.

3) To evaluate the effectiveness of the proposed algorithm, comparative experiments were conducted using three widely adopted recognition algorithms: the Rules recognition algorithm, the CNN recognition algorithm, and the DBSCAN clustering recognition algorithm. The results indicate that the MAPE (Mean Absolute Percentage Error) value of the proposed directed edge mining recognition algorithm is reduced by an average of 56%, 49%, and 32% compared to the aforementioned algorithms. This demonstrates that the proposed method achieves higher recognition accuracy and is more suitable for identifying regular traffic routes.

The main contribution of this study is to propose a regular route mining algorithm based on directed edges, which can effectively identify users' regular travel paths from GPS trajectories and improve the accuracy of traffic pattern recognition. For practical deployment, key challenges include the computational overhead for real-time processing of large-scale urban data, which is mitigated by our R-tree indexing achieving $O(|P|\log|P| + |R|^2)$ complexity, and the need for contextual calibration of thresholds like stop-rate across different urban environments. The specific value for traffic management systems lies in providing fine-grained, individual-level insights into recurrent travel patterns and mode choices, enabling more accurate dynamic traffic prediction at corridor levels by understanding traffic flow composition, and facilitating highly personalized navigation and smart travel services that account for users' habitual routes and preferred transport modes. These capabilities ultimately enhance urban traffic management efficiency and user travel experience beyond the limitations of aggregate-level models.

## Supporting information

**S1 File. Supporting information -figure-PLOS ONE.**
(DOCX)

**S2 File. Supporting information – lab data.**
(DOCX)

## Acknowledgments

Xiaobo Yang expresses gratitude to the scientific research team at Zhejiang Shuren University.

## Author contributions

**Methodology:** Xiaobo Yang.

**Writing – original draft:** Xiaobo Yang.

**Writing – review & editing:** Xiaobo Yang.

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
