## [Decision Letter · Decision Letter 0]

1 Sep 2025

Dear Dr. Yang,

Thank you for submitting your manuscript to PLOS ONE. After careful consideration, we feel that it has merit but does not fully meet PLOS ONE’s publication criteria as it currently stands. Therefore, we invite you to submit a revised version of the manuscript that addresses the points raised during the review process.

We look forward to receiving your revised manuscript.

Kind regards,

Guangyin Jin

Academic Editor

PLOS ONE

Journal Requirements:

4. We notice that your supplementary figures are uploaded with the file type 'Figure'. Please amend the file type to 'Supporting Information'. Please ensure that each Supporting Information file has a legend listed in the manuscript after the references list.

Reviewers' comments:

Reviewer's Responses to Questions

**Comments to the Author**

1. Is the manuscript technically sound, and do the data support the conclusions?

Reviewer #1: Partly

Reviewer #2: No

Reviewer #3: Partly

2. Has the statistical analysis been performed appropriately and rigorously?

Reviewer #1: No

Reviewer #2: No

Reviewer #3: No

3. Have the authors made all data underlying the findings in their manuscript fully available?

Reviewer #1: Yes

Reviewer #2: No

Reviewer #3: Yes

4. Is the manuscript presented in an intelligible fashion and written in standard English?

Reviewer #1: No

Reviewer #2: No

Reviewer #3: Yes

Reviewer #1: This paper proposes a direction aware conventional route mining algorithm for traffic pattern recognition based on travel trajectories. The research content has good application value. Before it can be published, the following problems need to be solved

1. The current title cannot reflect the core innovation of the research. It is suggested that the author further improve it.

2. The improvement of MAPE index mentioned in the abstract suggests adding specific reasons for choosing MAPE or introducing other more suitable evaluation indicators to enhance the credibility of the conclusion.

3. There are many abbreviations in the paper. It is recommended that the author carefully check to ensure that each abbreviation is given its corresponding full name when it first appears (such as GPS, CNN), and add a list of abbreviations.

4. The paper is not comprehensive enough in the analysis of the current research situation. The following related literature is suggested for review: https://doi.org/10.1016/j.physa.2023.128980, https://doi.org/10.1109/JSEN.2020.3007809; DOI: 10.1109/TITS.2023.3346473

5. The paper mentions setting the grid size to "10s" and draws the conclusion from Figure S3 that "the error distance is about 50m, which meets the resolution of urban roads". However, there is no specific definition of the "10s" grid size, such as whether it is a time parameter corresponding to the spatial grid edge length? Or the conversion logic between time grid and spatial grid? Furthermore, the selection criteria for this parameter were not specified.

6. It is recommended that the authors further improved to summarize the contributions from this study and supplement the analysis on the application of the research findings..

Reviewer #2: The abstract claims large performance improvements over rules, CNN, and DBSCAN approaches, citing MAPE reductions of 56%, 49%, and 32% respectively. Yet the evaluation metric is mismatched with the task. MAPE is appropriate for continuous regression tasks, not for discrete route mining or mode recognition. The absence of justification for using MAPE undermines the claimed contributions. In the introduction, prior work is cited in bulk without synthesis, and the contribution of the present study remains unclear. Instead of positioning the method against known baselines with a clear research gap, the introduction lists references but never builds a logical bridge explaining why direction-aware edges or stop-rates are fundamentally new or superior. The methods section is the weakest part of the paper. Nearly every stage of the algorithm depends on thresholds and parameters, but the actual values and rationales are missing. For example, minimum and maximum speed deviations, acceleration thresholds, stop distances and times, frequency and support thresholds, and trajectory similarity criteria are all introduced symbolically , yet no numerical values or tuning strategies are provided. Without them, the method cannot be reproduced.

Equations are also inconsistently formatted and at times corrupted with Unicode errors (e.g., “lengtℎ,” “tℎresℎold,” “folows”), which is unacceptable in a scientific manuscript. Subscripts and superscripts are inconsistently applied, and notation is inconsistent across sections. Key elements such as the map-matching procedure, which is crucial for trajectory-to-road distance calculation, are omitted entirely. The authors describe speed as “distance along the road network between GPS points,” but the details of network preprocessing, snapping, or error tolerance are absent.

The dataset description raises immediate concerns. The authors report that Geolife data spans 2010–2022, yet the well-known Geolife dataset was collected primarily between 2007 and 2012. This discrepancy suggests either a misunderstanding or careless reporting. Results claim that 428 “regular routes” were extracted, but the supporting visualizations are relegated to supplementary figures that are neither visible in the main text nor properly labeled. The figures included are of low resolution, poorly labeled, and in many cases relegated to supplementary materials. A reader cannot independently verify or appreciate the claimed results because axes, units, and legends are missing or incomplete. No tables summarize threshold values, performance metrics, or sensitivity analyses. PLOS ONE requires that figures central to the claims be part of the main text with full explanatory captions; here, the images appear to be an afterthought.

Reviewer #3: The work proposes a direction-aware regular-route mining algorithm that (i) denoises raw GPS trajectories, (ii) detects stay regions, (iii) clusters temporally aligned sub-trajectories, and (iv) extracts frequent directed edges and their supporting paths to reconstruct a user’s routine routes. A “stop-rate” feature derived from these routes is then used to discriminate public transit from private-car travel. Experiments on the GeoLife data set (208 users, 1.35 M km) report 56 %, 49 % and 32 % lower MAPE than rule-based, CNN and DBSCAN baselines, respectively. The paper claims usefulness for dynamic traffic prediction and personalized route recommendation. Several comments are as follows:

1) The four hyper-parameters ηs_d_min, ηs_d_max, ηaccelerate, ηd_stop, ηt_stop, ηt_split, ηfrequency, ηsupport, ηtrajectory, ηspeed_threshold and ηstop are introduced without justification or sensitivity analysis. Provide (i) data-driven or domain-knowledge-based derivation, (ii) grid-search or Bayesian optimisation results, and (iii) confidence intervals for all reported MAPE reductions under parameter perturbation.

2) The chosen CNN baseline (EfficientNetV2) is an image classifier; its adaptation to sequential GPS data is neither described nor referenced. Replace or supplement it with trajectory-specific deep models such as DeepMove, TrajCNN or ST-Transformer, and ensure identical feature sets and training protocols.

3) MAPE can be misleading when travel times are short. Include complementary metrics (F1-score for mode classification, edit-distance similarity for route extraction, Hausdorff distance for path geometry, and runtime/scalability) and analyse failure cases.

4) The literature review would benefit from the inclusion of more recent works on deep learning . Such as 10.1016/j.energy.2025.137730, 10.1007/s40747-024-01693-9, etc.

5) GeoLife is heavily skewed toward Beijing and single urban modality. Demonstrate generalisability on at least one additional city-scale data set (e.g., T-Drive, Porto taxi, or New York TLC). Discuss demographic biases (age, income, vehicle ownership) that may affect stop-rate assumptions.

6) The core novelty—“direction-aware directed-edge clustering”—lacks theoretical grounding. Provide complexity analysis (time & space), convergence proof for the iterative SE extraction, and an ablation study that disables each module (stay detection, directionality, stop-rate) to quantify individual contribution.

**Do you want your identity to be public for this peer review?** For information about this choice, including consent withdrawal, please see our Privacy Policy

Reviewer #1: No

Reviewer #2: No

Reviewer #3: No

---

## [Author Response · Author response to Decision Letter 1]

10 Sep 2025

Dear editor,

I have made revisions to the manuscript based on the review comments and submitted the relevant materials to the submission system for your review.

---

## [Decision Letter · Decision Letter 1]

24 Sep 2025

Dear Dr. Yang,

Thank you for submitting your manuscript to PLOS ONE. After careful consideration, we feel that it has merit but does not fully meet PLOS ONE’s publication criteria as it currently stands. Therefore, we invite you to submit a revised version of the manuscript that addresses the points raised during the review process.

We look forward to receiving your revised manuscript.

Kind regards,

Guangyin Jin

Academic Editor

PLOS ONE

Reviewers' comments:

Reviewer's Responses to Questions

**Comments to the Author**

Reviewer #1: (No Response)

Reviewer #3: All comments have been addressed

2. Is the manuscript technically sound, and do the data support the conclusions?

Reviewer #1: Partly

Reviewer #3: Yes

3. Has the statistical analysis been performed appropriately and rigorously?

Reviewer #1: N/A

Reviewer #3: Yes

4. Have the authors made all data underlying the findings in their manuscript fully available?

Reviewer #1: No

Reviewer #3: Yes

5. Is the manuscript presented in an intelligible fashion and written in standard English?

Reviewer #1: No

Reviewer #3: Yes

Reviewer #1: 1The research lacks theoretical comparisons and modular contribution analyses with existing advanced methods. It is suggested to quantitatively analyze the contribution degree of each module to the overall performance, and at the same time provide theoretical proofs of the algorithm's time complexity and space complexity.More details are provided in the attachment.

Reviewer #3: The authors have adequately addressed my previous comments and the manuscript is now suitable for publication. However, please note a discrepancy in Reference 22: the entry in the reference list is authored by **Chen**, while the in-text citation refers to **Huang** — please verify and correct either the in-text citation or the reference list entry so that they match and conform to the journal’s reference style. After this minor correction I recommend acceptance for publication.

**Do you want your identity to be public for this peer review?** For information about this choice, including consent withdrawal, please see our Privacy Policy

Reviewer #1: No

Reviewer #3: No

---

## [Author Response · Author response to Decision Letter 2]

21 Oct 2025

Dear editor,

According to the opinions of the reviewers, the manuscript has been revised again. Please refer to the materials uploaded by the system for details. Please review.

---

## [Decision Letter · Decision Letter 2]

30 Oct 2025

Dear Dr. Yang,

Thank you for submitting your manuscript to PLOS ONE. After careful consideration, we feel that it has merit but does not fully meet PLOS ONE’s publication criteria as it currently stands. Therefore, we invite you to submit a revised version of the manuscript that addresses the points raised during the review process.

We look forward to receiving your revised manuscript.

Kind regards,

Guangyin Jin

Academic Editor

PLOS ONE

Journal Requirements:

Reviewers' comments:

Reviewer's Responses to Questions

**Comments to the Author**

Reviewer #1: All comments have been addressed

Reviewer #3: (No Response)

2. Is the manuscript technically sound, and do the data support the conclusions?

Reviewer #1: Yes

Reviewer #3: Yes

3. Has the statistical analysis been performed appropriately and rigorously?

Reviewer #1: Yes

Reviewer #3: Yes

4. Have the authors made all data underlying the findings in their manuscript fully available?

Reviewer #1: Yes

Reviewer #3: Yes

5. Is the manuscript presented in an intelligible fashion and written in standard English?

Reviewer #1: Yes

Reviewer #3: Yes

Reviewer #1: My overall impression of this manuscript is very positive. Its most outstanding strength lies in its clear innovativeness, demonstrated through the authors' clever experimental design. Furthermore, the manuscript exhibits a logical structure, substantial data, a convincing argumentation process, and fluent writing.

Reviewer #3: 1. The mathematical symbols in the manuscript have not been properly formatted using the appropriate tools

2. Upon reviewing the manuscript, I found that reference [22] is incorrect. Both the title and authors are inaccurate. The correct article is titled “Predictive modeling and multi-objective optimization of magnetic core loss with activation function flexibly selected Kolmogorov-Arnold networks”, and its DOI is https://doi.org/10.1016/j.energy.2025.137730. Please use Google Scholar to locate the full citation in the proper format. Additionally, please ensure that the following paper is correctly cited: https://doi.org/10.1016/j.knosys.2025.114514. This paper is valuable for describing machine learning concepts.

**Do you want your identity to be public for this peer review?** For information about this choice, including consent withdrawal, please see our Privacy Policy

Reviewer #1: No

Reviewer #3: No

---

## [Author Response · Author response to Decision Letter 3]

24 Nov 2025

Dear reviewers,

According to the review comments, appropriate modifications have been made in the submission system. Please review it.

---

## [Decision Letter · Decision Letter 3]

30 Nov 2025

Directed-Edge-Based Mining of Regular Routes for Enhanced Traffic Pattern Recognition from Travel Trajectories

PONE-D-25-43817R3

Dear Dr. Yang,

We’re pleased to inform you that your manuscript has been judged scientifically suitable for publication and will be formally accepted for publication once it meets all outstanding technical requirements.

Kind regards,

Guangyin Jin

Academic Editor

PLOS ONE

Additional Editor Comments (optional):

Reviewers' comments:

Reviewer's Responses to Questions

**Comments to the Author**

Reviewer #3: All comments have been addressed

2. Is the manuscript technically sound, and do the data support the conclusions?

Reviewer #3: No

3. Has the statistical analysis been performed appropriately and rigorously?

Reviewer #3: Yes

4. Have the authors made all data underlying the findings in their manuscript fully available?

Reviewer #3: Yes

5. Is the manuscript presented in an intelligible fashion and written in standard English?

Reviewer #3: Yes

Reviewer #3: It was a great pleasure to review this manuscript. The reviewers have resolved all my issues, and the current form can be accepted.

**Do you want your identity to be public for this peer review?** For information about this choice, including consent withdrawal, please see our Privacy Policy

Reviewer #3: No

---

## [Editor Report · Acceptance letter]

PONE-D-25-43817R3

PLOS One

Dear Dr. Yang,

I'm pleased to inform you that your manuscript has been deemed suitable for publication in PLOS One. Congratulations! Your manuscript is now being handed over to our production team.

Kind regards,

on behalf of

Dr. Guangyin Jin

Academic Editor

PLOS One